# Pathophysiological Role of Microglial Activation Induced by Blood-Borne Proteins in Alzheimer’s Disease

**DOI:** 10.3390/biomedicines11051383

**Published:** 2023-05-07

**Authors:** Sehwan Kim, Chanchal Sharma, Un Ju Jung, Sang Ryong Kim

**Affiliations:** 1School of Life Sciences, Kyungpook National University, Daegu 41566, Republic of Korea; arputa@naver.com (S.K.); chanchalmrt@gmail.com (C.S.); 2Brain Science and Engineering Institute, Kyungpook National University, Daegu 41944, Republic of Korea; 3BK21 FOUR KNU Creative BioResearch Group, Kyungpook National University, Daegu 41566, Republic of Korea; 4Department of Food Science and Nutrition, Pukyong National University, Busan 48513, Republic of Korea; jungunju@naver.com

**Keywords:** Alzheimer’s disease, microglia, neuroinflammation, neurodegeneration, blood–brain barrier, blood-borne protein

## Abstract

The blood–brain barrier (BBB) restricts entry of neurotoxic plasma components, blood cells, and pathogens into the brain, leading to proper neuronal functioning. BBB impairment leads to blood-borne protein infiltration such as prothrombin, thrombin, prothrombin kringle-2, fibrinogen, fibrin, and other harmful substances. Thus, microglial activation and release of pro-inflammatory mediators commence, resulting in neuronal damage and leading to impaired cognition via neuroinflammatory responses, which are important features observed in the brain of Alzheimer’s disease (AD) patients. Moreover, these blood-borne proteins cluster with the amyloid beta plaque in the brain, exacerbating microglial activation, neuroinflammation, tau phosphorylation, and oxidative stress. These mechanisms work in concert and reinforce each other, contributing to the typical pathological changes in AD in the brain. Therefore, the identification of blood-borne proteins and the mechanisms involved in microglial activation and neuroinflammatory damage can be a promising therapeutic strategy for AD prevention. In this article, we review the current knowledge regarding the mechanisms of microglial activation-mediated neuroinflammation caused by the influx of blood-borne proteins into the brain via BBB disruption. Subsequently, the mechanisms of drugs that inhibit blood-borne proteins, as a potential therapeutic approach for AD, along with the limitations and potential challenges of these approaches, are also summarized.

## 1. Introduction

Alzheimer’s disease (AD) is a progressive neurodegenerative disorder often characterized by memory loss and impaired cognition [1,2,3,4]. The global AD prevalence is increasing concurrently with the aging population [4,5,6,7]. However, the pathogenic mechanisms of AD remain elusive, despite extensive research on the various causative and contributory factors, including genetics [1,2,3,8,9,10,11,12]. An accumulating body of evidence suggests that neuroinflammation mediated by glial activation is an important neurotoxic mechanism associated with the initiation and progression of AD, including the subsequent neuronal cell death and cognitive impairment [13,14,15,16,17,18,19,20,21,22,23,24].

Microglia promote brain homeostasis through synaptic remodeling maintenance and dead cell removal under non-pathological conditions. However, during the early stages of AD, an increase in inflammatory responses via microglial activation in the brain has been reported [23,25,26]. These responses are also involved in the early formation of amyloid beta (Aβ) plaques and persistent microglial activation, which can result in the generation of pro-inflammatory cytokines and reactive oxygen species (ROS) [23,25,26]. These pathways interact and reinforce one another to induce pathological alterations in the AD brain, which primarily involve microglial activation, neuroinflammation, tau hyperphosphorylation, compromised mitochondrial activity, and oxidative stress [23,25,26]. This, in turn, triggers neurodegeneration due to the subsequent loss of synapses and neurons, resulting in neuronal death [14,15,17,18,20]. In particular, the initial microglial activation and the resulting neuroinflammation may act as a starting point for the observed neurodegeneration in the AD brain [23,25,26]. These findings suggest that initial microglial activation can exacerbate AD progression, despite potential variations in the roles of microglia activation and inflammatory responses within the brain. [13,17,18,27,28,29,30]. Therefore, there is a pressing need to investigate such endogenous substances that may induce microglial activation to build an effective AD treatment strategy.

The blood–brain barrier (BBB) is essential for the maintenance of the brain tissue microenvironment. It improves connectivity while also providing insulation of the brain parenchyma from the peripheral circulation system [31]. It is composed of specialized endothelial cells and various tight junction (TJ) proteins to ensure optimal functioning of the BBB [31]. These TJs, together with specialized transporters, allow the passage of nutrients into the central nervous system (CNS) while restricting blood-borne molecule influx [32]. Meanwhile, in pathological conditions, BBB disruption can result in the entry of blood-borne proteins such as prothrombin, thrombin, prothrombin kringle-2 (pKr-2), fibrinogen, and fibrin, among others, into the brain parenchyma [33,34]. This causes gradual neurodegeneration and neuronal death through direct and/or indirect neurotoxicity, oxidative stress, mitochondrial function loss, and pro-inflammatory mediator release, leading to memory loss and a decline in cognitive function. Several reports suggest that the early stages of AD are critical, as BBB breakdown occurs during this period [35,36,37,38,39], which allows the entry of blood-borne proteins in the brain parenchyma, thus contributing to neurotoxicity by several mechanisms [40,41]. For instance, blood-borne proteins such as prothrombin, thrombin, pKr-2, fibrinogen, and fibrin can activate microglia and induce neuroinflammation, resulting in neurodegeneration and cognitive decline in various animal models of AD [34,42,43,44,45,46,47,48,49,50,51]. Moreover, some of these blood proteins (such as thrombin, fibrinogen, and fibrin) can cluster with Aβ, the main causal agent of AD, and can further exacerbate microglial activation [52,53,54,55,56]. Furthermore, fibrinogen deposition can lead to impaired memory and result in exacerbated levels of interleukin-6, ROS, mitochondrial superoxide, and nitrite in mouse brain neurons [57]. This can result in abnormal tau phosphorylation, which is another major hallmark of AD [58], and further amplify neurotoxicity through the interplay with Aβ plaques, thus affecting neurotransmitter balance [59]. As fibrinogen is not expressed in the brain, BBB disruption is required for plasma fibrinogen to cross the BBB and reach the brain parenchyma [34]. Therefore, investigating the mechanisms of blood-borne protein entry into the brain and microglial activation promotion leading to the release of pro-inflammatory cytokines resulting in neuronal loss may help develop novel therapeutic strategies for AD.

In this review, we first provide a brief overview of microglial activation and its transition into various phenotypes. The interplay between M1 (pro-inflammatory) and M2 (anti-inflammatory) phenotypes were also described by emphasizing their role in AD. Additionally, the pathophysiology and consequences of BBB disruption that further lead to neuronal loss and cognitive impairment observed during AD initiation and progression have been elucidated. Furthermore, we also describe the mechanisms and phenomena of microglial activation and neuroinflammation caused by the infiltration of various blood-borne proteins through the BBB. Subsequently, a summary of the mechanisms of drugs that inhibit blood-borne proteins, as a potential therapeutic approach for AD, along with the limitations and potential challenges of these approaches, is also discussed.

## 2. Microglial Activation in AD

As resident innate immune cells in the CNS, microglia are the primary regulators of brain inflammation [14,15,19,20,60]. There are numerous forms of activated microglia, some of which may be beneficial, while others may be harmful [14,15,16,20,29,61,62]. A growing body of evidence suggests that microglial activation in the CNS can be classified into two contrasting phenotypes: the pro-inflammatory M1 phenotype and the anti-inflammatory M2 phenotype (Figure 1) [63,64]. The classical activation state of microglia, known as M1 microglia, is characterized by the induction of inducible nitric oxide synthase (iNOS) and nuclear factor-κB (NF-κB) pathways, resulting in the production of various pro-inflammatory cytokines, including tumor necrosis factor-alpha (TNF-α), interleukin-1 beta (IL-1β), and IL-6, as well as superoxide, ROS, and nitric oxide (NO) [63,64]. On the other hand, M2 microglia encompass a spectrum of activation states induced by interleukin-4 (IL-4)/interleukin-13 (IL-13) and interleukin-10 (IL-10)/transforming growth factor-beta (TGF-β) signaling, which are collectively referred to as alternative activation and acquired deactivation, respectively [63,64]. The question of how microglia alter phenotypes when the underlying clinical insult is in the peripheral organs and systemic circulation is critical.

In the brain tissue of individuals with AD, microglia play a crucial role [14,15,17,18,19,27,29,62,65]. They participate in the brain’s inflammatory response and assist in the removal of brain debris and foreign substances [19,28,60,66]. For example, microglia in the brain of patients with AD remove Aβ and tau protein accumulation and promote the repair of damaged areas [19,28,60,66]. However, excessive activation of microglia in the brain of AD patients can also lead to neuroinflammation, which can accompany neurodegeneration [24,30,66]. Moreover, the excessive activation of microglia in the brain can cause neuronal damage, leading to memory loss and cognitive impairment [60,67,68]. In the early stages of AD plaque development, M2 microglia participate in Aβ phagocytosis by producing proteolytic enzymes that decrease Aβ production, thereby promoting tissue homeostasis [19,28,66,69]. However, with the excessive Aβ plaque accumulation in AD, M1 microglia continue to be activated, resulting in persistent inflammation and a vicious cycle of even greater Aβ production and neurodegeneration [13,17,18,27,28,29,30]. Moreover, microglia also prevent the spread of tau by internalizing and/or destroying the tau seeds [70]. However, the failure of microglia to degrade tau results in the release of exosomes containing tau that can spread to neurons, resulting in impaired mitochondrial activity, synaptic dysfunction, neurodegeneration, and cognitive decline [70]. Microglia have an extremely low mitochondrial turnover and critically suffer from mitochondrial dysfunction [71]. Furthermore, both in vitro and in vivo experiments suggest the role of microglia in the preservation of the BBB phenotype through the enhancement of TJ protein expression [72]. Conversely, emerging evidence advocates that microglial activation is associated with increased BBB permeability and a decrease in the clearance of toxic substances from the brain, which has been implicated in promoting inflammatory responses within the BBB [73,74]. Although the initiation of an innate immune response by microglia is aimed at protecting the brain, its overactivation may result in disease progression [24,30,75]. Therefore, understanding the mechanisms that activate microglia is crucial.

Neuroinflammation is a pathological hallmark of AD [22,23,24,25,75,76,77], and, while it rarely occurs in a normal brain, it occurs in pathologically vulnerable areas. Activated microglia and astrocytes produce inflammatory factors such as cytokines, chemokines [Monocyte chemoattractant protein-1 (MCP-1), interleukin-8 (IL-8)], NO, and various complement factors such as complement component 1q (C1q), complement component 3 (C3), complement component 4 (C4) [14,29,77,78,79,80,81]. Neuroinflammation secondary to early microglial activation in AD can impair other brain cell functions, cause neuronal damage and BBB leakage, and accelerate AD progression [13,17,18,27,28,29,30]. Unlike other risk factors for AD, neuroinflammation is usually not only a causative factor itself but rather a result of one or more AD pathologies (e.g., Aβ, hyperphosphorylated tau) or endogenous factors associated with AD [69,79,82]. It also appears to have a dual function that plays a neuroprotective role during the acute phase reaction but can be harmful under chronic conditions; persistently activated microglia can cause neuronal damage by releasing a variety of pro-inflammatory and toxic products [25,30,75,77,78]. According to several reports, inflammatory responses by microglial activation in the brain increase in the early stages of AD [23,25,26]. Persistent activation of microglia can lead to the production of pro-inflammatory cytokines and ROS, which may be the starting point for the neurodegeneration observed in the AD brain. [23,25,26]. Therefore, investigating endogenous substances that activate microglia is essential for identifying the causes of this microglial activation in the early stages of AD.

## 3. Pathophysiology of BBB Disruption

The BBB plays a crucial role in maintaining CNS homeostasis [83,84,85,86]. It is composed of endothelial cells, astrocyte end feet, and pericytes, which form a highly specialized structure that restricts the passage of blood-borne substances into the CNS (Figure 2) [31,83,87,88]. The TJs between endothelial cells form the primary barrier, preventing the movement of hydrophilic molecules and macromolecules between the blood and the brain parenchyma, thus playing a critical role in regulating the paracellular transport of molecules and ions between endothelial cells (Figure 2A) [31,87,88,89,90,91]. The astrocyte endfeet are intimately associated with the endothelial cells and secrete factors that regulate the BBB structure and function [31,87,88]. Pericytes, located on the abluminal side of the endothelial cells, play a crucial role in BBB maintenance and regulation [31,87,88]. Collectively, these cellular components work in consort to maintain the integrity of the BBB and regulate the selective transport of nutrients, ions, and signaling molecules to the CNS, while preventing the entry of potentially harmful substances (Figure 2B) [92,93,94].

TJs are composed of transmembrane proteins, such as occludin and claudins, as well as intracellular scaffolding proteins, such as the zonula occludin family proteins (Figure 2A) [89,90,91]. The interplay of these proteins forms a tight seal between adjacent endothelial cells, preventing the diffusion of large molecules and maintaining the selective permeability of the BBB [89,90,91,92]. Dysregulation or breakdown of TJs can lead to increased permeability of the BBB, allowing for the influx of harmful substances such as neurotoxic blood-borne products (such as prothrombin, thrombin, fibrinogen, fibrin, pKr-2, albumin), toxic aggregated proteins (such as Aβ and tau), autoantibodies, iron, sodium, cells, and pathogens into the brain, thereby initiating inflammatory and immune responses and contributing to the development of several neurodegenerative diseases [40,41,95].

Several clinical and animal studies suggest that disruption of BBB function plays a critical role in the initiation and progression of AD [37,96,97,98,99,100,101,102,103,104,105,106]. An increased influx of blood-borne proteins and neurotoxic substances into the brain induces neuroinflammatory responses that can potentially exacerbate AD pathology [35,36,37,97,99,100,101,107]. Additionally, it has been suggested that the passage of complement proteins across the BBB alters the function of neurons, oligodendrocytes, and microglia; permits the infiltration of inflammatory cells into the brain; and triggers cytokine cascades that further aid in AD progression [35,36,37]. This suggests the possibility that the neuroinflammation observed in the early stages of AD may be via microglial activation [23,25,26], which may be related to the BBB breakdown and subsequent entry of blood-borne proteins into the brain [35,36,97]. Hence, it is important to investigate the mechanisms of microglia activation by endogenous molecules that enter through BBB leakage during the development of AD.

## 4. Blood-Borne Protein by BBB Leakage Induces Microglial Activation in AD

The BBB prevents the entry of harmful substances into the brain from the blood [72,88,108,109]. Age-related degeneration of the BBB increases its permeability along with a decrease in cerebral blood flow [95,109,110,111,112]. A decrease in the interaction between neurons and endothelial cells and astrocytes is another factor that contributes to the BBB breakdown and impairment. This results in the blood-borne substance leakage into the brain, including prothrombin, thrombin, pKr-2, fibrinogen, fibrin, and albumin (Figure 3) [34,46,51,86].

### 4.1. Prothrombin and Thrombin

Prothrombin is a protein involved in the blood coagulation pathway, which helps to form blood clots in response to injury [113,114,115,116,117]. Prothrombin in blood plasma is converted into thrombin by factor X or prothrombinase, which then transforms fibrinogen into fibrin to form a clot with platelets, preventing blood loss [113,114,115,116,117]. Increased expression of prothrombin has been found in both neurons and glial cells in the brain tissue of patients with AD patients compared with healthy controls [48,50,118,119,120]. Furthermore, prothrombin has been shown to accumulate in neurofibrillary tangles, a pathological hallmark of AD [48]. Although the precise mechanisms underlying the upregulation of prothrombin and thrombin in AD are not completely understood, a growing body of evidence suggests their potential role in the pathogenesis of this neurodegenerative disease [48,49,50,51,118,119]. These findings suggest that prothrombin may contribute to the neurodegenerative process in AD, particularly concerning neuronal damage [48,118,120]. Thrombin is a protease that is also involved in the blood coagulation pathway, but it can also be generated in the brain in response to injury and inflammation [121,122,123]. In AD, thrombin is upregulated in the brain and may contribute to the pathogenesis of the disease [48,50,118]. In one study, the expression of prothrombin mRNA in brain tissues, neuroblastoma cells, cultured human astrocytes, oligodendrocytes, and microglial cells were detected using the reverse transcriptase–polymerase chain reaction, and the presence of prothrombin and thrombin proteins in the brain was confirmed using specific monoclonal and polyclonal antibodies for both proteins through immunohistochemistry [48]. Furthermore, studies have demonstrated increased expression of thrombin and its receptors such as protease-activated receptor 1 (PAR-1) and protease-activated receptor 4 (PAR-4) in the brain in neurodegenerative diseases [118,124]. Thrombin has also been reported to induce the NADPH oxidase in microglia in the hippocampus, leading to the death of neuronal cells, suggesting a potential mechanism by which thrombin contributes to neurodegenerative processes in the brain [49]. Other reports also show that thrombin can activate microglia, leading to pro-inflammatory cytokine releases such as TNF-α and IL-1β in BV-2 and primary human microglia in vitro [125,126].

Thrombin-induced neuroinflammation via microglial activation was found to be mediated through the PAR-1 receptor signaling pathway [124]. In the AD brain, thrombin can also induce vascular inflammation, exacerbating BBB damage [50,51]. Thrombin may also contribute to Aβ accumulation in the brain, as it can cleave the Aβ precursor protein and generate Aβ peptides that aggregate and form amyloid plaques in the cultured hippocampal neurons and human endothelial cells in vitro [52,53]. Additionally, reports suggest that thrombin can induce tau hyperphosphorylation and aggregation in murine hippocampal neurons, resulting in microglial activation, reduction in delayed synaptophysin, and apoptotic neuronal death [127]. Moreover, along with calpain-1, it can cleave tau, resulting in a ~17 kDA neurotoxic fragment that can cause impaired synaptic function and axonal transport, mitochondrial membrane depolarization, and behavioral deficits in AD brains [128,129].

### 4.2. pKr-2

pKr-2, also known as fragment 2, is a small protein fragment that is released during the activation of prothrombin [44,45,46,47,130,131,132,133,134,135,136,137,138,139,140,141,142], which is a precursor protein that is converted into the active blood-clotting enzyme thrombin [44,45,46,47,130,131,132,133,134,135,136,137,138,139,140,141,142]. The role of pKr-2 in the blood is to inhibit the prothrombinase complex activity in conjunction with Factor Va, thereby preventing further thrombin production [45,139,143]. In addition, pKr-2 inhibits endothelial cell proliferation and angiogenesis, suggesting its potential role in BBB maintenance [135,136].

We recently reported the increased pKr-2 protein expression in the brain tissues of patients with AD and 5XFAD AD mouse models compared to those of healthy individuals using Western blotting and immunofluorescence staining [44,45,46]. In an in vitro study, the pKr-2 induced neuroinflammation through microglial activation without directly causing neuronal toxicity in the co-culture of mesencephalic neurons and microglia [134]. However, in the hippocampus of the 5XFAD AD mouse model, pKr-2 overexpression was shown to induce microglial activation, leading to excessive neuroinflammation and subsequent neuronal death through the activation of TLR4 transcription factors, such as PU.1 and p-c-Jun [44,45,47,130,133]. We also reported that caffeine and rivaroxaban treatment can suppress the generation and brain influx of pKr-2, leading to a reduction in neuroinflammation and improvement in cognitive impairment in the 5XFAD AD mouse model [46]. Although further detailed mechanistic studies are required to fully understand the microglial activation by pKr-2, various studies have demonstrated that controlling pKr-2 overexpression is a potential therapeutic strategy for AD [44,45,46,47].

### 4.3. Fibrinogen, Fibrin

Fibrinogen is a key protein involved in the blood-clotting process [34,42,43,54,144,145,146]. The formation of fibrin monomers at the site of vascular injury induces the binding of fibrinogen to the vessel wall and its conversion into fibrin dimers, which form a foam-like structure in the blood and are important for blood clotting [144,145,147]. Recent studies have shown that fibrinogen can traverse the BBB and have significant physiological and pathological effects in the brain [34,42,43,54]. Fibrinogen is associated with neuroinflammation and changes in brain vasculature in neurodegenerative disorders such as AD [42,43]. These changes can cause damage to brain neurons and synapses, leading to symptoms such as cognitive impairment [34]. According to the in vivo study by Merlini et al., fibrinogen can activate microglia in the brain and contribute to the process of removal of synaptic spines, which is associated with the development and progression of AD in mice [34]. Additionally, fibrinogen can impair the integrity of the brain vasculature and cause BBB damage, which occurs in the early stages of AD and increases in severity as the disease progresses [43,146]. Another study suggested that the binding of fibrinogen and Aβ in the brain may exacerbate the severity of a TgCRND8 AD model in vivo [54,55,56], as this association can result in the formation of abnormal blood clots and blockages in the cerebral vasculature [54].

Similar to fibrinogen, fibrin also plays an important role in the process of blood clotting [42,144,145,147,148]. Thrombin cleaves fibrinogen to form fibrin monomers, which polymerize to create the fibrin clot [42,144,145,147,148]. According to a study, conducted in APP/PS1 transgenic mice, an AD model, an increase in fibrin within blood vessels and the abnormal accumulation of the extracellular matrix in the brain tissue were found to induce neurovascular damage and neuroinflammatory responses [148]. The progression of Aβ deposition in the brain tissue was accompanied by vascular damage and inflammatory responses [148], which, in turn, accelerated the progression of AD due to the excessive accumulation of fibrin [148]. Thus, reducing the abnormal accumulation of fibrin in the extracellular matrix and brain vasculature may mitigate brain damage and slow down disease progression [148]. Ryu et al. showed that treatment with the 5B8 antibody targeting the fibrin epitope γ377-395 prevented abnormal fibrin accumulation in a 5XFAD AD animal model, resulting in decreased neuroinflammation and neuronal damage [42].

### 4.4. Other Proteins and Factors

In addition to coagulation factors, blood vessels contain a diverse array of proteins and factors that serve a range of functions beyond their role in hemostasis [83,84,85,86,149,150,151,152,153,154,155]. These proteins and factors can also infiltrate the brain and have harmful effects [149,150,151,152,153,154,155]. In one study, the blood-borne Aβ protein was found to penetrate the BBB and accumulate in the microglia, leading to AD pathogenesis [155]. The blood Aβ protein induces neuroinflammation and microglial activation, leading to the formation of brain deposits and neuronal damage [155]. AD patients have increased levels of albumin and immunoglobulin G (IgG) in their blood compared to normal individuals, and the passage of these proteins through the impaired BBB into the brain may play a role in the pathophysiology of AD [149,151,154]. Additionally, the brain tissues of patients with AD patients were shown to have increased concentration of hemoglobin-derived peptides, which may contribute to neuroinflammation and neuronal damage [153]. Another study reported increased iron levels in the hippocampal region of patients with AD using magnetic resonance imaging, which may be related to the pathophysiological changes in AD [152]. Plasminogen derived from blood was shown to induce Aβ accumulation in the brain through microglial activation, further aggravating the neuroinflammatory responses [150]. Specifically, plasminogens derived from blood could induce Aβ accumulation in the brain through microglial activation, leading to further promotion of neuroinflammatory responses [150]. Therefore, further investigation is required to examine additional proteins and factors present in the blood and to investigate their mechanisms of infiltration into the brain and activation of microglial cells.

In addition, the overexpression or overactivation of certain brain proteins could disrupt BBB integrity. A study investigated the role of IL-1β in regulating BBB permeability and found that IL-1β activated microglia, leading to damage and inflammation of vascular endothelial cells; this, subsequently, affected angiogenesis and increased vascular permeability [156]. In addition, an investigation of BBB destruction and cell death in an ischemic stroke model revealed that the TNF-α derived from microglia induced neuronal death, exacerbating BBB destruction [157]. Some studies revealed that blood-derived extracellular vesicles, such as microvesicles, transported into the brain could influence microglial activation, and extracellular vesicles generated through microglial activation could induce neuronal damage in the AD brain [158,159]. Moreover, High Mobility Group Box-1 (HMGB1) was identified as one of the key factors that promotes BBB breakdown. HMGB1 increases in the setting of brain injury or inflammation, inducing an inflammatory response in the endothelial cells of the BBB and increasing BBB permeability [160]. Matrix metalloproteinases (MMPs) are protein-degrading enzymes that regulate the extracellular matrix, but excessive MMP activity can cause brain tissue damage and inflammation [160,161]. Increased activity of MMP-9 was shown to be involved in the breakdown of the BBB [161]. Therefore, further research on the signaling pathways such as MMP-9 and other proteins in relation to BBB damage may support the development of new therapeutic strategies for AD.

## 5. Novel Therapeutic Approach to the Control of Blood-Borne Proteins in AD

Anticoagulants are used as preventive or therapeutic measures against blood-borne proteins, thereby impeding thrombosis or embolism formation [113,162]. These drugs inhibit blood clotting through various mechanisms [113,162]. Several studies have reported the potential role of anticoagulants that can inhibit blood-borne proteins in the treatment of AD (Figure 4).

Heparin is an anticoagulant that is primarily administered as an intravenous or subcutaneous injection to treat thrombotic disorders such as acute myocardial infarction and pulmonary embolism [163,164,165,166,167,168,169,170]. Several studies have reported that heparin could increase gamma-secretase activity and decrease beta-secretase 1 (BACE1) activity in primary cortical neurons from Tg2576 mice, leading to the inhibition of Aβ production [164]; it could also regulate Apolipoprotein E (ApoE), a protein that binds to Aβ, leading to reduced Aβ generation, neurotoxicity, and brain inflammation [165,167,168]. Furthermore, heparin has been reported to inhibit microglial activation and inflammation in the brain, which could be beneficial for treating AD [165,166]. Enoxaparin, a low molecular weight heparin, was shown to inhibit Aβ deposition and improve cognitive function in the Appel/PS1dE9 mouse model of AD, suggesting its potential role in the treatment of AD [169]. In addition, enoxaparin was found to prevent the formation of ApoE4-Aβ complexes by binding to them [170]. In another study, dalteparin, a low molecular weight heparin, was found to reduce oxidative stress and inhibit the inflammatory response in the hippocampus of the streptozotocin-induced model, leading to an improvement in cognitive impairment [171].

Dabigatran is an oral anticoagulant that acts by directly inhibiting thrombin [172,173,174,175,176]. The administration of dabigatran for 12 months in the TgCRND8 AD mouse model led to the reduced accumulation of Aβ aggregates and a decrease in neuronal damage and inflammatory response [175]. In another study using a PS19 mouse model with tau aggregates, the expression pattern of some proteins related to mitochondria was altered after a one-week treatment with dabigatran [174]. In addition, in primary rat cortical neurons, dabigatran treatment resulted in a decrease in thrombin-induced pro-inflammatory cytokine and chemokine expression, as well as Aβ peptide generation, and a decrease in the level of phosphorylated tau protein [173]. Furthermore, a study suggested that dabigatran may prevent the progression of AD by inhibiting inflammation, protecting the brain vasculature by inhibiting thrombin activity [172]. These results indicated that dabigatran may reduce neuroinflammation and AD pathology related to thrombin activity.

Rivaroxaban is an anticoagulant that directly inhibits factor Xa associated with thrombin production [177,178,179]. An earlier report suggested that a high dose of rivaroxaban (60 mg/kg) can attenuate neuroinflammation, BBB dysfunction, memory deficits, and Aβ deposition through PAR-1/PAR-2 inhibition in the AD CAA mice model [180]. Recently, we also reported that oral administration of rivaroxaban to six-month-old 5XFAD mice for three months inhibited the production of pKr-2 in the hippocampus, resulting in reduced neuroinflammation, neurotoxicity, and improved object cognitive behavior [46].

The available evidence suggests that anticoagulants have sufficient potential to suppress neuroinflammation mediated via microglial activation by inhibiting blood-borne proteins in the brain [46,163,164,165,166,167,168,169,170,172,173,174,175,176]. Although nonsteroidal anti-inflammatory drugs, such as cyclooxygenase 1 and 2 inhibitors, can be used to control neuroinflammation in the AD brain [181,182], several studies have suggested that these anticoagulants can ameliorate neuroinflammation through microglial activation inhibition, thereby impeding blood-derived proteins and Aβ generation and aggregation [165,166,167,168,169,170,173,175].

## 6. Conclusions

As AD is a multifactorial disorder, with various mechanisms contributing to its progression and development, it cannot be concluded that the disease progression occurs solely due to specific mechanisms of blood-borne proteins. Moreover, further research is required for an in-depth characterization of the neuroinflammatory mechanisms associated with microglial activation induced by blood-borne protein entry through the impaired BBB. Altogether, these studies suggest that abnormal blood coagulation and the proteins involved in the neuroinflammatory process play a central role in the development and progression of AD. Additionally, the deterrent effects against blood-borne proteins may be a potential therapeutic strategy in the early stage of AD.

## Figures and Tables

**Figure 1 biomedicines-11-01383-f001:**
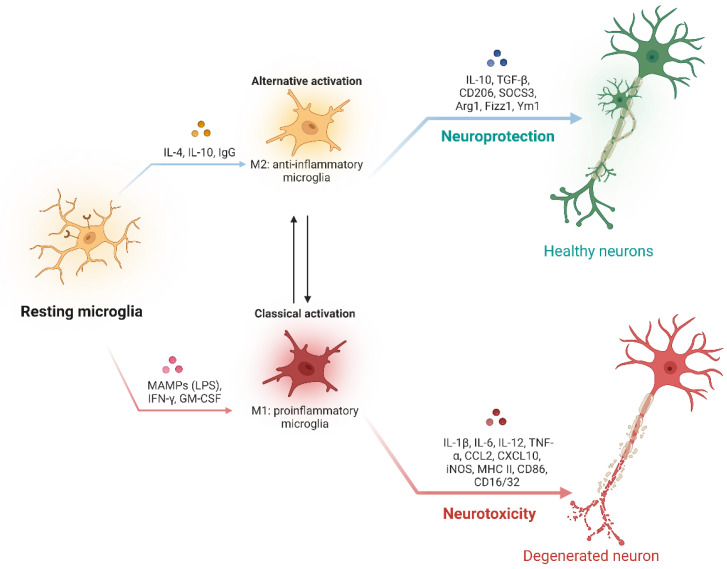
Schematic illustration of the polarization and phenotypic transition of microglia. Microglia remain stationary in a normal state. Various stimuli can induce the resting microglia to acquire different phenotypes. In general, M1 microglia is a pro-inflammatory phenotype, while M2 microglia is an anti-inflammatory phenotype. M1 or M2 microglia can release different substances, and these two phenotypes are interchangeable under certain conditions. Figure was created using BioRender (https://www.biorender.com/ (accessed on 5 April 2023); Agreement number: RT257J0VDX).

**Figure 2 biomedicines-11-01383-f002:**
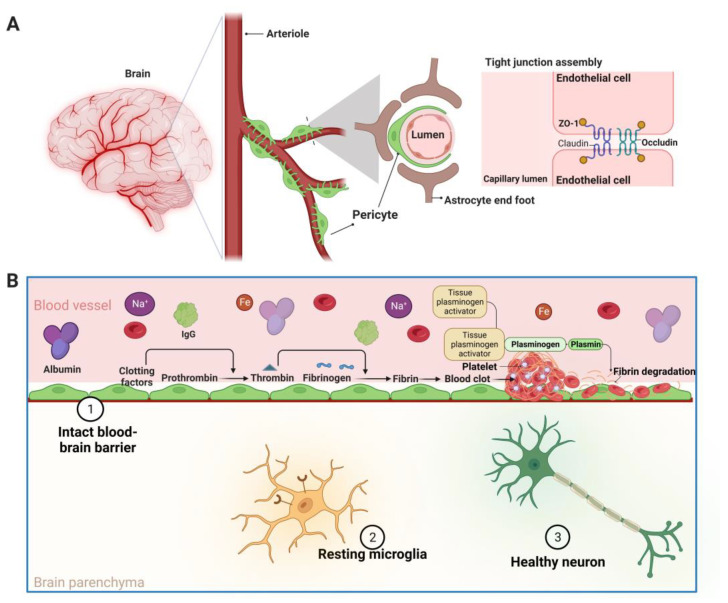
The blood–brain barrier (BBB) in the healthy brain. (**A**) The BBB consists of a complex cellular system of a highly specialized basal membrane, a large number of pericytes, astrocytic end feet, tight junction proteins (occludin and claudin), neurons, and microglia. The cells forming the BBB communicate with cells of the brain and in the periphery. The pericytes and endothelium share a common basement membrane and connect with a variety of transmembrane junction proteins, which aid in BBB integrity maintenance. Pericytes, endothelial cells, and neurons are connected by astrocytes. The immunological responses are regulated by microglia. (**B**) The BBB regulates the blood-to-brain and brain-to-blood permeation of several substances, resulting in the nourishment of the CNS, its homeostatic regulation, and communication between the CNS and peripheral tissues. Figure was created using BioRender (https://www.biorender.com/ (accessed on 6 May 2023); Agreement number: WK25BZA2TM).

**Figure 3 biomedicines-11-01383-f003:**
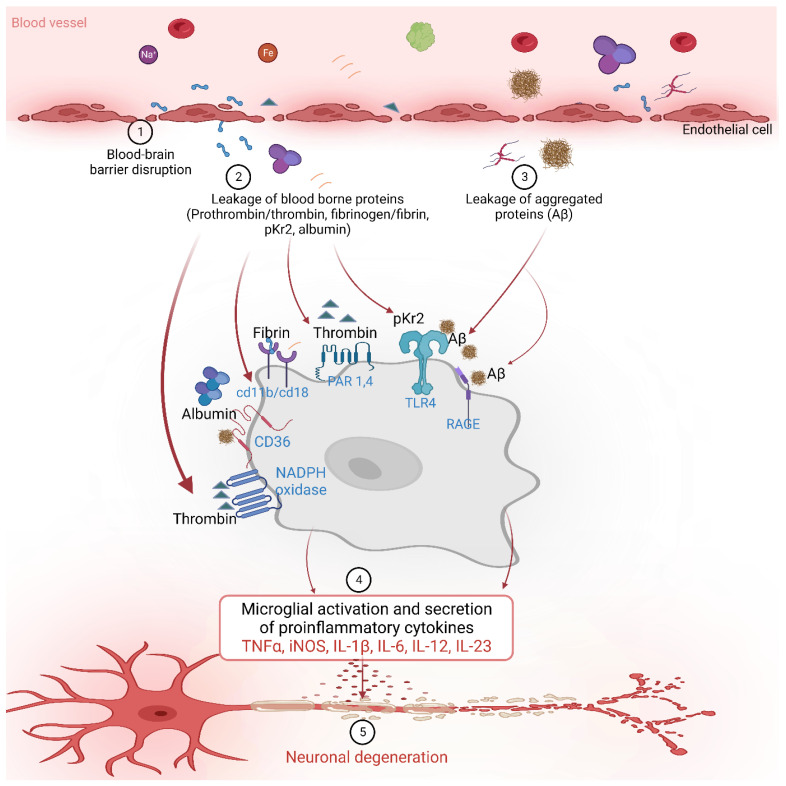
The blood-borne proteins interact with various receptors on the microglial cells to activate downstream signaling and influence various inflammatory and neurodegenerative processes in Alzheimer’s disease. Figure was created using BioRender (https://www.biorender.com/ (accessed on 5 April 2023); Agreement number: ST257J0OKO).

**Figure 4 biomedicines-11-01383-f004:**
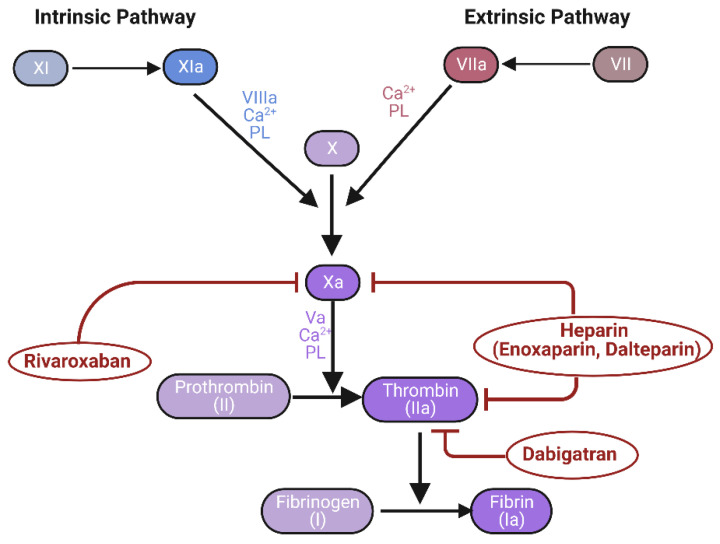
Targets of various anticoagulants in the coagulation cascade. Figure was created using BioRender (https://www.biorender.com/ (accessed on 5 April 2023); Agreement number: NB257IZC0E).

## Data Availability

Not applicable.

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
