# Peer review of "Pathophysiological Role of Microglial Activation Induced by Blood-Borne Proteins in Alzheimer’s Disease"

_biomedicines, 2023, doi:10.3390/biomedicines11051383_

Round 1

Reviewer 1 Report

Paper looks good and these are minor concerns needs to be fixed.

1.      Lack of clear research question or objective: The abstract presents a review of current knowledge about the mechanisms of microglial activation and blood-brain barrier disruption in Alzheimer's disease, but it's not entirely clear what the authors are trying to accomplish with this review. Are they proposing a new hypothesis? Are they evaluating the efficacy of different therapeutic strategies? Without a clear research question or objective, it's hard for readers to know what to expect from the article.

2.       Lack of context: While the abstract mentions several key features of Alzheimer's disease (microglial activation, BBB disruption, and blood-borne proteins), it doesn't provide much context for why these features are important or what their broader implications are. For example, how do these features relate to the cognitive decline and memory loss that are characteristic of Alzheimer's disease? Without this context, it's difficult for readers to fully understand the significance of the research.

3.       Lack of specificity: The abstract mentions several different blood-borne proteins that are involved in microglial activation and neuroinflammation, but it doesn't provide much detail about how these proteins work or how they contribute to the development of Alzheimer's disease. Similarly, the abstract mentions "potential strategies" for controlling these proteins, but it doesn't specify what these strategies are or how effective they might be. More specificity would help readers better understand the scope of the research and its potential implications.

4.       Possible oversimplification: The abstract suggests that regulating blood-borne proteins is a promising therapeutic strategy for Alzheimer's disease, but it's not clear how feasible or effective this approach might be. Alzheimer's disease is a complex and multifaceted condition, and it's unlikely that a single therapeutic strategy will be sufficient to address all its underlying causes. While targeting blood-borne proteins may be a promising avenue of research, it's important to acknowledge the limitations and potential challenges of this approach.

5.       Lack of focus: The introduction covers a broad range of topics related to Alzheimer's disease, including its prevalence, neuroinflammation, oxidative stress, microglial activation, and blood-brain barrier disruption. While all of these topics are relevant to the discussion of AD, the introduction does not clearly state a specific focus or research question.

6.       Lack of clarity: The introduction contains a significant amount of technical jargon and complex sentence structures that may be difficult for non-experts to understand. It could benefit from clearer language and simpler sentence structures to make it more accessible to a broader audience.

7.       Incomplete review of the literature: While the introduction cites a large number of references, it does not provide a comprehensive review of the relevant literature. For example, it does not discuss the many other factors that have been implicated in the development and progression of AD, such as tau protein aggregation, mitochondrial dysfunction, and neurotransmitter imbalances.

8.       Weak thesis statement: The introduction does not clearly state the author's thesis or research question. A strong thesis statement would provide a clear focus for the paper and guide the reader through the rest of the content.

9.       The text cites numerous references to support its claims, but it does not always provide specific details about the methods or results of the studies mentioned. This makes it difficult to evaluate the quality or relevance of the evidence presented.

10.   The text presents a simplified view of microglial activation as either M1 or M2 phenotypes, which may not fully capture the complexity of microglial responses in different contexts. Moreover, the distinction between M1 and M2 phenotypes has been challenged by recent research that suggests a more nuanced and dynamic interplay between pro-inflammatory and anti-inflammatory signaling pathways in microglia.

11.   The text suggests that neuroinflammation is a secondary or downstream effect of other AD pathologies, but this may not always be the case. Some studies have proposed that microglial activation and neuroinflammation can contribute to the initiation and progression of AD pathology, rather than being a consequence of it.

12.   The text focuses mainly on the detrimental effects of microglial activation and neuroinflammation, but it does not fully acknowledge the potential benefits of these processes in certain circumstances. For example, microglia may play a beneficial role in clearing Aβ plaques or promoting tissue repair, and neuroinflammation may be a necessary component of the brain's immune response to injury or infection.

English is adequate

Author Response

Response to Reviewer 1:

Reviewer #1: Paper looks good and these are minor concerns needs to be fixed.

Response: We thank the reviewers for the thorough review of our manuscript and providing constructive comments and suggestions for manuscript revision.

1. Comment:Lack of clear research question or objective: The abstract presents a review of current knowledge about the mechanisms of microglial activation and blood-brain barrier disruption in Alzheimer's disease, but it's not entirely clear what the authors are trying to accomplish with this review. Are they proposing a new hypothesis? Are they evaluating the efficacy of different therapeutic strategies? Without a clear research question or objective, it's hard for readers to know what to expect from the article.

Response: We appreciate the reviewer’s comment. Accumulating evidence suggests that glial activation-mediated neuroinflammation is an important neurotoxic mechanism associated with neurodegeneration and cognitive impairment observed in AD brains. Limited evidence exists indicating the role of blood-borne proteins capable of inducing neurodegeneration and impaired cognition via neuroinflammatory responses through microglial activation. Hence, in this review, a summary of the blood-borne proteins and their mechanisms involved in microglial activation and neuroinflammatory damage in AD brains was discussed. We have summarized the drug mechanisms that can inhibit blood-borne proteins, as a potential therapeutic approach for AD, along with the limitations and potential challenges of these approaches.

As suggested, text revision was done accordingly to ensure clarity (Abstract and Section 1: Introduction).

2. Comment: Lack of context: While the abstract mentions several key features of Alzheimer's disease (microglial activation, BBB disruption, and blood-borne proteins), it doesn't provide much context for why these features are important or what their broader implications are. For example, how do these features relate to the cognitive decline and memory loss that are characteristic of Alzheimer's disease? Without this context, it's difficult for readers to fully understand the significance of the research.

Response: We thank the reviewer for their insightful comment. As suggested, the text has been updated concerning the importance and implications of microglial activation, BBB disruption, and blood-borne proteins in AD-associated cognitive decline and memory loss (Sections 1 and 2).

3. Comment: Lack of specificity: The abstract mentions several different blood-borne proteins that are involved in microglial activation and neuroinflammation, but it doesn't provide much detail about how these proteins work or how they contribute to the development of Alzheimer's disease. Similarly, the abstract mentions "potential strategies" for controlling these proteins, but it doesn't specify what these strategies are or how effective they might be. More specificity would help readers better understand the scope of the research and its potential implications.

Response: We appreciate the reviewer’s comment. As suggested, we have briefly explained the mechanism of these blood-borne proteins and their contribution to AD development in Section 4, and Figure 3. Potential strategies to control blood-borne protein-mediated microglial activation and neuroinflammation have been reviewed in Section 5 and Figure 4.

4. Comment: Possible oversimplification: The abstract suggests that regulating blood-borne proteins is a promising therapeutic strategy for Alzheimer's disease, but it's not clear how feasible or effective this approach might be. Alzheimer's disease is a complex and multifaceted condition, and it's unlikely that a single therapeutic strategy will be sufficient to address all its underlying causes. While targeting blood-borne proteins may be a promising avenue of research, it's important to acknowledge the limitations and potential challenges of this approach.

Response: We thank the reviewer for the suggestions. We concur that AD is a multifactorial disease and various mechanisms contribute to its progression and development. As suggested, the limitations of current approaches and potential challenges are included in the revised manuscript under “Section 6.”

5. Comment: Lack of focus: The introduction covers a broad range of topics related to Alzheimer's disease, including its prevalence, neuroinflammation, oxidative stress, microglial activation, and blood-brain barrier disruption. While all of these topics are relevant to the discussion of AD, the introduction does not clearly state a specific focus or research question.

Response: We thank the reviewer for the insightful comments. As suggested, the Introduction section has been revised to clarify the research objective of the current manuscript.

6. Comment:Lack of clarity: The introduction contains a significant amount of technical jargon and complex sentence structures that may be difficult for non-experts to understand. It could benefit from clearer language and simpler sentence structures to make it more accessible to a broader audience.

Response: We thank the reviewer for the comment. As advised, we have simplified the sentences to improve readability.

7. Comment: Incomplete review of the literature: While the introduction cites a large number of references, it does not provide a comprehensive review of the relevant literature. For example, it does not discuss the many other factors that have been implicated in the development and progression of AD, such as tau protein aggregation, mitochondrial dysfunction, and neurotransmitter imbalances.

Response: We appreciate the reviewer’s comment. We have included a discussion on the key findings of the cited references accordingly. Also, factors contributing to AD pathogenesis were also elucidated.

8. Comment: Weak thesis statement: The introduction does not clearly state the author's thesis or research question. A strong thesis statement would provide a clear focus for the paper and guide the reader through the rest of the content.

Response: We thank the reviewer for their insightful comments. As indicated, the Introduction section has been revised accordingly to improve clarity regarding the study’s purpose.

9. Comment: The text cites numerous references to support its claims, but it does not always provide specific details about the methods or results of the studies mentioned. This makes it difficult to evaluate the quality or relevance of the evidence presented.

Response: We thank the reviewer for the valuable comment. As suggested, the manuscript was revised accordingly concerning the methods and results of cited references to ensure clarity.

10. Comment: The text presents a simplified view of microglial activation as either M1 or M2 phenotypes, which may not fully capture the complexity of microglial responses in different contexts. Moreover, the distinction between M1 and M2 phenotypes has been challenged by recent research that suggests a more nuanced and dynamic interplay between pro-inflammatory and anti-inflammatory signaling pathways in microglia.

Response: We thank the reviewer for the suggestion. We agree that microglia are important brain cells that support brain homeostasis through synaptic remodeling, maintenance, and dead cell removal under non-pathologic conditions. However, CNS inflammation caused by activated microglial cells is one of the key features of various neurodegenerative diseases, including Parkinson’s disease (PD) and AD (Choi et al., 2005; Kim et al., 2010, 2021; Shin et al., 2015; Chung et al., 2020). When stimulated by activators, microglial cells undergo phagocytic morphological changes indicated by enlarged cell bodies and short processes. Invading pathogens promote microglial cell activation, producing pro-inflammatory mediators including various cytokines, chemokines, inducible nitric oxide synthase, and cyclooxygenase 2, ultimately leading to neurotoxicity and neurodegenerative disease progression (Choi et al., 2005; Kim et al., 2010, 2021; Chung et al., 2020). Additionally, activated microglial cells can also produce reactive oxygen species which are also involved in neuroinflammatory processes causing neurodegeneration and further cooperatively drive the pathology observed in the AD brain. Inflammation activates microglia and polarizes them into the M1 (cytotoxic) or M2 (neuroprotective) phenotype; thus, microglia can either harm or protect the BBB. Altogether, it is essential to examine the endogenous molecules and pathogenic mechanisms associated with microglial activation that cause neurotoxic brain inflammation to develop a useful treatment strategy for AD.

The interplay between the phenotypic transition of microglia during AD initiation and progression via several mechanisms such as Aβ plaque accumulation, tau hyperphosphorylation, and BBB impairment has been elucidated in Section 2.

11. Comment: The text suggests that neuroinflammation is a secondary or downstream effect of other AD pathologies, but this may not always be the case. Some studies have proposed that microglial activation and neuroinflammation can contribute to the initiation and progression of AD pathology, rather than being a consequence of it.

Response: We appreciate the reviewer’s comment. Inflammation is essentially an organism’s defensive response. However, excessive and dysregulated inflammation results in negative consequences. Various cytokines, including IL-1β, IL-6, IL-9, IL-17, interferon-γ, TNF-α, and CCL2, can reduce the expression of tight junction proteins, responsible for the selective permeability of the BBB. Because tight junctions are one of the most critical BBB components, any change in the tight junction protein expression directly impacts the BBB resulting in the influx of various toxins into the brain parenchyma, further exacerbating neuroinflammation, and increasing neurodegenerative disease risk.

However, this paper focused on reviewing blood-borne protein-induced microglial activation and corresponding neuroinflammatory damage due to BBB impairment, which is one of the primary characteristics observed in the AD brain (Nation et al., 2019; van de Haar et al., 2016; Ujiie et al., 2003; Kawas, 2003; Henneman et al., 2009; Xiao et al., 2020; Desai et al, 2007). Hence, the manuscript suggests that neuroinflammation is a secondary or downstream effect of other AD pathologies such as BBB disruption (Section 2).

12. Comment: The text focuses mainly on the detrimental effects of microglial activation and neuroinflammation, but it does not fully acknowledge the potential benefits of these processes in certain circumstances. For example, microglia may play a beneficial role in clearing Aβ plaques or promoting tissue repair, and neuroinflammation may be a necessary component of the brain's immune response to injury or infection.

Response: We thank the reviewer for the insightful comment. As suggested, text revision was employed concerning the potential benefits of microglial activation in Sections 1 and 2 of the manuscript.

References:

Choi, S.H.; Lee, D.Y.; Kim, S.U.; Jin, B.K. Thrombin-induced oxidative stress contributes to the death of hippocampal neurons in vivo: role of microglial NADPH oxidase. J Neurosci 2005, 25, 4082-4090, doi:10.1523/JNEUROSCI.4306-04.2005.

Chung, Y.C.; Jeong, J.Y.; Jin, B.K. Interleukin-4-Mediated Oxidative Stress Is Harmful to Hippocampal Neurons of Prothrombin Kringle-2-Lesioned Rat In Vivo. Antioxidants (Basel) 2020, 9, doi:10.3390/antiox9111068.

Desai, B.S.; Monahan, A.J.; Carvey, P.M.; Hendey, B. Blood-brain barrier pathology in Alzheimer's and Parkinson's disease: implications for drug therapy. Cell Transplant 2007, 16, 285-299, doi:10.3727/000000007783464731.

Henneman, W.J.; Sluimer, J.D.; Barnes, J.; van der Flier, W.M.; Sluimer, I.C.; Fox, N.C.; Scheltens, P.; Vrenken, H.; Barkhof, F. Hippocampal atrophy rates in Alzheimer disease: added value over whole brain volume measures. Neurology 2009, 72, 999-1007, doi:10.1212/01.wnl.0000344568.09360.31.

Kawas, C.H. Clinical practice. Early Alzheimer's disease. N Engl J Med 2003, 349, 1056-1063, doi:10.1056/NEJMcp022295.

Kim, S.R.; Chung, E.S.; Bok, E.; Baik, H.H.; Chung, Y.C.; Won, S.Y.; Joe, E.; Kim, T.H.; Kim, S.S.; Jin, M.Y.; et al. Prothrombin kringle-2 induces death of mesencephalic dopaminergic neurons in vivo and in vitro via microglial activation. J Neurosci Res 2010, 88, 1537-1548, doi:10.1002/jnr.22318.

Nation, D.A.; Sweeney, M.D.; Montagne, A.; Sagare, A.P.; D'Orazio, L.M.; Pachicano, M.; Sepehrband, F.; Nelson, A.R.; Buennagel, D.P.; Harrington, M.G.; et al. Blood-brain barrier breakdown is an early biomarker of human cognitive dysfunction. Nat Med 2019, 25, 270-276, doi:10.1038/s41591-018-0297-y.

Shin, W.H.; Jeon, M.T.; Leem, E.; Won, S.Y.; Jeong, K.H.; Park, S.J.; McLean, C.; Lee, S.J.; Jin, B.K.; Jung, U.J.; et al. Induction of microglial toll-like receptor 4 by prothrombin kringle-2: a potential pathogenic mechanism in Parkinson's disease. Sci Rep 2015, 5, 14764, doi:10.1038/srep14764.

Ujiie, M.; Dickstein, D.L.; Carlow, D.A.; Jefferies, W.A. Blood-brain barrier permeability precedes senile plaque formation in an Alzheimer disease model. Microcirculation 2003, 10, 463-470, doi:10.1038/sj.mn.7800212.

van de Haar, H.J.; Burgmans, S.; Jansen, J.F.; van Osch, M.J.; van Buchem, M.A.; Muller, M.; Hofman, P.A.; Verhey, F.R.; Backes, W.H. Blood-Brain Barrier Leakage in Patients with Early Alzheimer Disease. Radiology 2016, 281, 527-535, doi:10.1148/radiol.2016152244.

Xiao, M.; Xiao, Z.J.; Yang, B.; Lan, Z.; Fang, F. Blood-Brain Barrier: More Contributor to Disruption of Central Nervous System Homeostasis Than Victim in Neurological Disorders. Front Neurosci 2020, 14, 764, doi:10.3389/fnins.2020.00764.

Reviewer 2 Report

The review under revision entitled “Pathophysiological role of microglial activation induced by blood-borne proteins in Alzheimer's disease” is interesting and deals with a topic characterized by a certain element of novelty. Is a good manuscript for which I have only few observations.

- Please check Author name. There seems to be some inaccuracy.

- Please check out typos. For example, blood-borne should be standardized along the manuscript. Somewhere appear blood-born.

- 4.4. Other proteins and factors section: have the author revised also the involvement of blood derived extracellular vesicles or microvesicles on microglia activation? Some information should be added.

-Novel therapeutic approach to the control of blood-born proteins in AD section: it would be interesting for the authors to discuss in this paragraph on the role of using cox-1 and cox-2 inhibitors against blood-borne proteins for the purpose of containing microglial activation and thus neuroinflammation that may be responsible for many neurodegenerative diseases including AD.

Overall, this manuscript has good potential and therefore can be considered for a possible publication after minor revision.

The quality of English is good.

Author Response

Response to Reviewer 2:

Reviewer #2: The review under revision entitled “Pathophysiological role of microglial activation induced by blood-borne proteins in Alzheimer's disease” is interesting and deals with a topic characterized by a certain element of novelty. Is a good manuscript for which I have only few observations.

1. Comment: Please check the Author name. There seems to be some inaccuracy.

Response: We thank the reviewer for pointing this out. The error has been rectified accordingly.

2. Comment: Please check out typos. For example, blood-borne should be standardized along the manuscript. Somewhere appear blood-born.

Response: We thank the reviewer for highlighting this. We have thoroughly checked the manuscript, and revisions were done to ensure consistency throughout the manuscript.

3. Comment: Other proteins and factors section: have the author revised also the involvement of blood derived extracellular vesicles or microvesicles on microglia activation? Some information should be added.

Response: We thank the reviewer for the valuable comment. To further enrich the text, the role of blood-derived extracellular vesicles or microvesicles on microglia activation with ample evidence is discussed in Section 4.4.

4. Comment: Novel therapeutic approach to the control of blood-born proteins in AD section: it would be interesting for the authors to discuss in this paragraph on the role of using cox-1 and cox-2 inhibitors against blood-borne proteins for the purpose of containing microglial activation and thus neuroinflammation that may be responsible for many neurodegenerative diseases including AD.

Response: As per recommendation, the role of cox-1 and cox-2 inhibitors has been incorporated in Section 5 of the revised manuscript.

5. Comment: Overall, this manuscript has good potential and therefore can be considered for a possible publication after minor revision.

Response: We thank the reviewers for the thorough review of our manuscript and for providing constructive comments and suggestions for manuscript revision.